# Microplastic Interactions and Possible Combined Biological Effects in Antarctic Marine Ecosystems

**DOI:** 10.3390/ani13010162

**Published:** 2022-12-31

**Authors:** Roberto Bargagli, Emilia Rota

**Affiliations:** Department of Physics, Earth and Environmental Sciences, University of Siena, Via P.A. Mattioli 4, IT-53100 Siena, Italy

**Keywords:** Antarctic marine biota, anthropogenic contaminants, climate change, cumulative stress, fish, krill, microplastics, penguins, polar skua, seals, zoobenthos

## Abstract

**Simple Summary:**

Plastic pollution is spreading worldwide and a growing number of reports point to the presence of plastic waste and microplastics in Antarctic marine ecosystems. Although the available data do not yet allow us to define the distribution of microplastics in the biotic and abiotic components of the Southern Ocean, in discussing the possible interactions with other contaminants in wastewater from scientific stations and their possible combined effects on primary producers and food webs, this review emphasizes the urgent need for standardized protocols of sampling and analysis of microplastics. Considering the unique oceanographic and biological characteristics of the Southern Ocean, we also suggest evaluating the likely cumulative stresses in Antarctic marine organisms and ecosystems due to exposure to climate-induced environmental changes, such as the recent decrease in sea-ice formation and seawater acidification.

**Abstract:**

Antarctica and the Southern Ocean are the most remote regions on Earth, and their quite pristine environmental conditions are increasingly threatened by local scientific, tourism and fishing activities and long-range transport of persistent anthropogenic contaminants from lower latitudes. Plastic debris has become one of the most pervasive and ubiquitous synthetic wastes in the global environment, and even at some coastal Antarctic sites it is the most common and enduring evidence of past and recent human activities. Despite the growing scientific interest in the occurrence of microplastics (MPs) in the Antarctic environment, the lack of standardized methodologies for the collection, analysis and assessment of sample contamination in the field and in the lab does not allow us to establish their bioavailability and potential impact. Overall, most of the Southern Ocean appears to be little-affected by plastic contamination, with the exception of some coastal marine ecosystems impacted by wastewater from scientific stations and tourist vessels or by local fishing activities. Microplastics have been detected in sediments, benthic organisms, Antarctic krill and fish, but there is no clear evidence of their transfer to seabirds and marine mammals. Therefore, we suggest directing future research towards standardization of methodologies, focusing attention on nanoplastics (which probably represent the greatest biological risks) and considering the interactions of MPs with macro- and microalgae (especially sea-ice algae) and the formation of epiplastic communities. In coastal ecosystems directly impacted by human activities, the combined exposure to paint chips, metals, persistent organic pollutants (POPs), contaminants of emerging interest (CEI) and pathogenic microorganisms represents a potential danger for marine organisms. Moreover, the Southern Ocean is very sensitive to water acidification and has shown a remarkable decrease in sea-ice formation in recent years. These climate-related stresses could reduce the resilience of Antarctic marine organisms, increasing the impact of anthropogenic contaminants and pathogenic microorganisms.

## 1. Introduction

Antarctica is the most remote and the last major wilderness on Earth and has been designated by the Antarctic Treaty’s Protocol on Environmental Protection (ATPEP; signed in Madrid in 1991 and entered into force in 1998) as a “nature reserve, devoted to peace and science”. Geographical isolation (enhanced by the Antarctic Circumpolar Current and the Antarctic cyclonic vortex), the lower human population density in the Southern Hemisphere and environmental management (ATPEP’s annex III and IV) have contributed to relatively pristine conditions in Antarctica and the Southern Ocean. However, dichlorodiphenyltrichloroethane (DDT) and its congeners have been detected in marine organisms in the Southern Ocean since 1966 [1] and the discovery in the late 1970s that chlorofluorocarbons (CFCs), mainly produced and used in the Northern Hemisphere, were involved in the recurrent formation of the “ozone hole” over Antarctica [2,3], showed that this remote and isolated region is inextricably linked to global processes and the impact of persistent contaminants released on other continents [4,5,6]. Soon after DDT, many other persistent chemicals, such as polychlorinated biphenyls (PCBs), were detected in the Antarctic environment and biota [7,8,9,10,11,12]. In the 1990s it was understood why many Persistent Organic Pollutants (POPs), neither produced nor applied in Antarctica, can reach the polar regions. Chemicals such as DDT, PCBs and halogenated aromatic compounds are semi-volatile and with repeated volatilization–condensation–deposition cycles move along temperature gradients, from tropical and temperate regions to progressively colder latitudes, where they condense and settle (cold distillation) [7]. The low temperature does not allow for further volatilization and reduces the degradation rate of organic compounds, making the polar regions a permanent sink of POPs. Since the promulgation of the Stockholm Conventions, the production and use of several POPs have been progressively banned, but these compounds will continue to settle and persist for decades in Antarctica. Furthermore, global warming, with melting of ice and thawing of permafrost soils, will remobilize contaminants deposited in the past, increasing their bioavailability for the Antarctic biota [13,14].

The Southern Ocean biota includes many endemic species with unique ecophysiological adaptations, resulting from the extreme abiotic conditions and the long-lasting geographic isolation [15,16,17,18]. Due to the strong seasonality of productivity, many animals store lipids as energy reserves, have poor metabolism of POPs and are prone to the accumulation of lipophilic contaminants and their biomagnification through a unique marine food chain based largely on Antarctic krill. Thus, in the tissues of Antarctic organisms at higher trophic levels and with a relatively long lifespan such as the south polar skua, POP concentrations may be close to those causing adverse health effects [19,20].

Over the past 70 years there has been a notable increase in the human presence in Antarctica, particularly on the Antarctic Peninsula, the South Shetland Islands and other islands just south of the Polar Front, such as South Georgia. In the austral summer 2019–2020, for instance, there were over 74,000 tourists and 5000 people in scientific stations [21,22]. Moreover, fairly intensive fishing is practised in the sub-Antarctic seas (north of 60° S, but still within the cold waters of the Antarctic Circumpolar Current). South of 60° S all human activities are regulated by ATPEP, but it is inevitable that, through the combustion of fuel (for transport and energy production), accidental oil spills, sewage and waste production, humans can generate local contamination hotspots [4,5,23,24,25,26,27]. Most scientific stations are situated on ice-free coastal areas, and in the context of the total coastline, surface and volume of the Southern Ocean (about 30,000 km, 22 million km^2^, and 71.8 million km^3^, respectively), their impact would seem significant only on a local scale. However, in these hotspots of anthropogenic disturbance, the concomitant impacts of contaminants, climate and environmental changes and the possible introduction of pathogenic microorganisms and/or invasive and pre-adapted alien species can pose cumulative stresses and threats to the biodiversity and functioning of marine ecosystems [28,29,30]. Furthermore, chemical production has continued to increase exponentially in recent decades and, while taking steps to reduce the use and impact of legacy pollutants, concern is growing over the global spread of microplastics (MPs; fragments with a diameter of <5 mm) and many synthetic chemicals not yet subject to regulatory criteria (the so-called Contaminants of Emerging Interest, CEI). Plastic litter and CEI have become a major focus of environmental research worldwide, and a rapidly growing number of surveys reports their occurrence in Antarctic marine and terrestrial ecosystems, especially those impacted by recent or past scientific activities [31,32,33,34,35]. The occurrence of plastic particles in the stomachs of seabirds nesting in Antarctica and the entanglement of seals in plastic debris have been documented for forty years [36,37]. However, only in the last decade, after MP pollution has taken on global environmental significance, has there been a proliferation of scientific research and reports on the occurrence of MPs in Antarctic sea-ice, waters, sediments and marine and land organisms [38,39]. In 2018, MPs were recognized by the Scientific Committee on Antarctic Research (SCAR) as a “serious emerging threat”, and the attention of the Antarctic Treaty System (ATS) to the issue and possible actions to prevent environmental contamination by plastic debris began to increase [40,41,42,43]. The ATS governs the entire region south of 60° S and since 1961 several Antarctic Specially Protected Areas (ASPAs) have been established to protect sites with outstanding natural, historic, environmental or wilderness values. Although these areas can be visited with permission and only for scientific activities that do not jeopardize the protected values, contamination by plastic debris has also been reported in some ASPAs [44,45].

Despite the growing interest, the data acquired so far on the distribution of MPs and their potential impact in Antarctic marine ecosystems are still scarce and refer mainly to areas with coastal scientific stations and higher anthropogenic pressures, such as the Antarctic Peninsula, the South Shetland Islands, those in the Scotia Arc and the Ross Sea. In addition, different methods for sampling, quantification and characterization of MPs were used and it is difficult to compare the results of different studies. The task is made even more difficult by the heterogeneity of the polymer types and the lack of knowledge on their degradation in the cold waters of the Southern Ocean (about −1.8 °C), with high ultraviolet (UV) radiation in summer and seasonal sea-ice cover. The Southern Ocean is characterized by very dynamic conditions in which seasonal pulses of sea-ice freezing and break-up influence the development of phytoplankton, ice-algae and algae growing on the underside of the pack ice, which can play an important role in the uptake of MPs by a key species in the food web such as Antarctic krill (*Euphausia superba*) [4,16,30]. Climate change can also constitute an additional stress factor through the warming and freshening of coastal waters and above all through the acidification of seawater, the Southern Ocean being among the seas most vulnerable to increasing CO_2_ concentrations [46]. The Antarctic continental shelf is about three times deeper than that of all the other seas and is home to well-developed benthic communities with a number of species probably second only to those of coral reefs [18]. In the marine environment, MPs can adsorb heavy metals, hydrophobic organic contaminants and pathogenic microorganisms and may also release plastic additives such as the antimicrobial triclosan, the antioxidant nonylphenol, or polybrominated diphenyl ethers (PBDEs; which confer thermo-resistance) [47,48,49,50,51]. Being distributed in sea-ice, surface waters, water column and bottom sediments, MPs with adsorbed contaminants and microorganisms can be ingested by cryopelagic, pelagic and benthic organisms belonging to all trophic levels. Changes in the local composition of biotic communities have been reported in coastal ecosystems affected by long-term anthropogenic pollution [20,23,26,29,52]. Moreover, benthic communities can play an important role in the transfer of MPs and other contaminants to fish, nesting seabirds and seals, and their involvement lengthens food webs and enhances the biomagnification of contaminants in top predators.

The possible combined effects of MPs and associated contaminants on marine organisms are becoming a research hotspot and although the role of MPs as vectors, relative to other routes of exposure, is largely unknown, a preliminary assessment seems particularly useful in the most-impacted Antarctic coastal ecosystems. Concentrations of MPs, POPs and CEI in these ecosystems are generally lower or in the same range as those in other seas; however, we cannot exclude possible biological effects in coastal biotic communities exposed to cumulative stresses due to interactions between MPs and other anthropogenic contaminants and the interplay of climatic and environmental changes. For example, the presence of MPs has been shown to increase the bioaccumulation and toxicity of cadmium (Cd) in zebrafish [53]. In Antarctica there are no rivers carrying to the sea major and trace elements leached from rocks and soils, and the development of phytoplankton can be limited by the poor bioavailability of major and trace elements such as silicon (Si) or zinc (Zn). In Southern Ocean waters, Cd has a similar distribution pattern to that of phosphate and behaves like a nutrient: diatoms take Cd atoms as a substitute for Zn to synthesize carbonic anhydrase (a metalloenzyme that provides carbon for photosynthesis) [54]. Thus, the metal is transferred from primary producers to consumers, and most Antarctic marine organisms accumulate high concentrations of Cd in the kidney and liver (or the digestive gland of invertebrates). In Antarctic seabirds and marine mammals, Cd concentrations often exceeded the values known to have toxic effects on other terrestrial birds or mammals, including humans [55,56,57]. Probably, during their evolution in the Southern Ocean, organisms have acquired metabolic pathways to detoxify Cd, but we cannot exclude that MPs may contribute to the bioaccumulation of Cd at levels above the threshold tolerance limit for this metal.

In summarizing the available data on the occurrence of MPs in Antarctic marine ecosystems and their potential effects on organisms and food webs with unique characteristics on Earth, this review will mainly focus on the interactions of MPs with other anthropogenic contaminants and on possible cumulative stress due to predicted changes in the Antarctic climate and marine environment. Human activities of greatest impact and the most vulnerable species and ecosystems will be highlighted, with the aim of guiding future research and contributing to the implementation of the ATPEP environmental management protocols.

## 2. Sources of Macro- and Microplastics in the Southern Ocean and Antarctica

When the ATPEP was adopted, the potential environmental impact of MPs, personal care products, pharmaceuticals, detergents, flame retardants and other persistent contaminants was not yet well understood, and wastewater from scientific stations and vessels operating in the Southern Ocean were generally considered to be biodegradable. On the other hand, although there were some reports of the ingestion of plastic debris by seabirds, these events seemed to mainly affect animals wintering outside the Antarctic [37], and the occurrence of beached macro- and mesoplastic fragments (>5 mm) had been observed mostly on the islands of the Scotia Arc, most impacted by fishing and other human activities [36,58,59]. Moreover, Lagrangian particle transport models indicate that coastlines and coastal waters are important reservoirs of marine plastic debris and, after 5 years, most of the buoyant fragments released from land-based sources become stranded or float in coastal water [60].

Long-term monitoring with standardized methodologies is required to evaluate the distribution of macro- and mesoplastics beached in the sub-Antarctic and to make comparisons between the different islands, since the results are influenced by large spatial and temporal variations in local weather and sea conditions, tide height, beach dynamics and variation in the durability and buoyancy of different groups of polymeric materials. Eriksson et al. [61], for instance, found that the accumulation rate of debris based on daily collection can be about an order of magnitude different than that from monthly collections. However, through a three-decade survey in the northern Scotia Sea, Waluda et al. [59] found a large prevalence of beached plastic debris (>5 mm) compared to glass, metal, paper or rubber debris; at Signy Island (southern Scotia Sea and within the ATS area) the monthly accumulation rate of beached debris was much lower than at more northern sites, which are more affected by fishing activities. Five-year monitoring (1990–1995) of beached debris on Bird Island (South Georgia) indicated that most of the plastic fragments came from longline fishing for the Patagonian toothfish *Dissosticus elaginoides* [44].

Although the available literature data from the Scotia Arc archipelagos shows that many beached plastic debris such as fishing-related materials or packaging bands are released by local or regional sources, there are also indications of plastic debris transported long distances by ocean currents [62,63]. In the past it was believed that the Antarctic Polar Front (APF, i.e., the boundary between the cold Antarctic and warmer sub-Antarctic waters) was an efficient circumpolar physico-chemical and biogeographic barrier, preventing the southward transport of floating or submerged materials. However, more recent oceanographic data indicate that the APF is not a continuous circumpolar jet and mesoscale eddies can carry sub-Antarctic water parcels southward [64,65]. Through a multi-year survey it has been found that two species of buoyant kelp that provide habitat for invertebrates can easily cross the APF and enter the Southern Ocean [66]. Therefore, even plastic debris with positive buoyancy and chemical stability can be transported southward, likely contributing to the transport of adsorbed persistent contaminants and alien marine organisms into the Southern Ocean.

Depending on the polymer composition, plastic litter can persist in the marine environment for decades to centuries, and exposure to ultraviolet light, waves, heat and microorganisms favors its degradation and the formation of tiny pieces (secondary MPs). Meanwhile weathering processes modify its physicochemical properties, promoting the release of dyes and plasticizing additives and the formation of functionalized oxygen moieties on their surfaces [67,68,69]. In sub-Antarctic seas, secondary and primary MPs can concentrate in deep waters due to wind-driven turbulence and vertical mixing [70] and can likely cross the APF. When they re-emerge in the surface waters of the Southern Ocean, they will probably disperse through the gyres of the Weddell and Ross Seas and the Antarctic Coastal Current. In addition, smaller MPs with micro to nano sizes can leave ocean waters at lower latitudes through bubble burst ejection and spume and will eventually reach the Southern Ocean via long-range atmospheric transport. Deposition of MPs (especially fibers) coming from great distances has been reported in other remote regions [71,72,73,74] and can also occur in the Antarctic environment. Analyzing snow samples collected up to 20 km away from McMurdo and Scott Base research stations, Aves et al. [75] found MPs (mainly fibers) in all samples (mean = 29 particles L^−1^). Retracing the trajectories of the air mass from the moment of sampling, they identified the items released by local sources (mainly polymers used in clothing and station equipment) and those deposited by snow after potential long-range transport. Assuming an atmospheric residence time of the fibers of 6.5 days, they estimated that the potential sources were located up to 6000 km away. According to Evangeliou et al. [76], global atmospheric emissions of MPs and microfibers are about 9.6 ± 3.6 Tg year^−1^ and 6.5 ± 2.9 Tg year^−1^, respectively; like POP compounds, the smallest particles can undergo a series of depositions followed by resuspension and are dispersed by wind all around the Earth. Their deposition in Antarctica is unknown; however, possible effects on the snow/ice surface albedos that favor the melting of the cryosphere and the release of contaminants deposited in the past cannot be excluded [77]. Airborne MPs can also influence Earth’s climate by absorbing and scattering solar radiation and acting as ice nuclei in clouds [78].

## 3. Microplastics in the Southern Ocean

The available data on MP concentrations in the Antarctic environment are scarce and patchy and it is worth mentioning once again that detailed analyses or comparisons between them are made difficult by different sampling, different analytical approaches and the lack of universally accepted methods for assessing MP contamination of samples in the field and in the lab. Moreover, although there is evidence that the smallest particles are more abundant and pose greater biological risks, the identification of small secondary MPs is still extraordinarily elusive and most surveys only consider those sized >300 µm. Pakhomova et al. [79], for instance, in reporting the concentrations of floating MPs in subsurface waters collected from the Arctic down to the Scotia Sea (south of 40°S), emphasize that the results can be very variable even in studies performed in the same year and in the same sea area and with similar methodological approaches. They found statistically significantly (*p* < 0.05) lower concentrations of MPs (>100 µm) in Southern than in Northern Hemisphere waters, but no significant differences between mean MP concentrations in the Barents Sea, the Central Atlantic, the Siberian Arctic, the North Atlantic and the sub-Antarctic (Table 1). In the Southern Hemisphere the content of fibers (synthetic or non-synthetic) was much lower and their features indicated local maritime activities as the main source, while in the Northern Hemisphere a large proportion of fibers originated from laundry processes and land-based wastewater treatment plants. These authors suggested that the lower percentage of fibers in the Central Atlantic compared to the Scotia Sea and the North Atlantic may be due to the rising of warm air masses at low latitudes, while in polar regions the descent of the cold air should favour the deposition of long-transported MPs.

Unlike all the other seas, the Southern Ocean has no rivers carrying MPs from land-based sources and available data (Table 1) indicate that the most-impacted areas are those affected by waters from coastal scientific stations and/or by fairly intensive scientific, tourism and fishing activities such as South Georgia Island, King George Island and the Ross Sea [33,79,80,81,82,83,84,85,86,87,88,89,90,91]. In general, in surface and subsurface waters of the Weddell Sea and in open waters of the Southern Ocean, MP concentrations are lower, and Kuklinski et al. [80] found that, although collected fibers appeared to be plastic, Fourier-Transform Infrared spectroscopy (FT-IR) analysis indicated that all fibers were composed of silica, suggesting a possible biogenic origin from diatoms (Table 1). However, long-range marine and atmospheric transport can contribute to MP occurrence in the Southern Ocean [31,33,35,39,83], and, as happens in other oceans, plastic debris should accumulate in the center of gyres. Measured concentrations of fragments sized >300 µm in surface and subsurface waters of the Weddell Sea gyre were much lower than those in the Arctic and the Atlantic (Table 1). However, these comparisons have only an indicative value; during two expeditions to the Weddell Sea, Leistenschneider et al. [83] found great temporal variability in the results and detected no MPs in water samples from coastal polynyas in the eastern Antarctic Peninsula. These coastal marine areas are free of sea-ice in winter as continental katabatic winds push coastal waters offshore and MPs are likely incorporated into newly-formed sea-ice [92].

A distinctive feature of the neuston samples from Southern Ocean waters is the high content of paint fragments released by research vessels [33,83,84]. Like plastic debris, paint fragments can adsorb contaminants and, if ingested, can also release metals and toxic organic compounds associated with paints and dyes. Some of these chemicals have already been detected in phytoplankton samples collected at Port Foster Bay (Deception Island) [93]. Moreover, paint debris has a rather high density and its settlement in marine sediments can be favored by biofouling or by attachment to faecal pellets and sea snow. Therefore, as shown by Muller-Karanassos et al. [94], it can also affect benthic organisms. Surveys on paint debris and MPs in offshore sediments of the Southern Ocean are lacking. A study on deep-sea sediment cores collected off the Antarctic Peninsula, South Sandwich Islands and South Georgia (i.e., the Antarctic marine areas more affected by scientific, tourism and fishing activities) shows that almost all samples contain MPs with a fairly homogeneous distribution in the three marine areas [95]. Mean concentrations of MPs (ranging from 1.04 ± 0.39 to 1.301 ± 0.51 items g^−1^) were higher than those reported in other less-remote sedimentary environments.

While common polymers from single-use products are widespread in most seas, in the Southern Ocean many items originate from the weathering and fragmentation of larger plastic objects and probably, there are also significant amount of plastic items of few-micrometer or sub-micrometer size (nanoplastics) deposited after long-range atmospheric transport. A recent survey reports higher concentrations of nanoplastics in a sea-ice core from Antarctica (52.3 ng mL^−1^) than in Greenland firn samples (13.2 ng mL^−1^) [96]. Polyethylene, polypropylene and polyethylene terephthalate were identified in Antarctic sea-ice nanoparticles, while in samples from Greenland firn, polystyrene, polyvinyl chloride and tyre-wear fragments were also recorded.

The distribution and bioavailability of MPs in the Southern Ocean is affected by complex processes such as marine and atmospheric circulation and seasonal formation and melting of sea-ice. Our knowledge of the behaviour of MPs at the seawater/ice interface is poor. However, Materic et al. [96] found more nanoplastics in Antarctic sea-ice than in seawater. Lab experiments to investigate the fate of micro- and nanoplastics during sea-ice formation indicate that microplastics are retained in ice, while nanoplastics are expelled from it and can be stabilized by natural organic matter at the seawater/ice interface [97]. This finding could have important implications for the absorption of plastics by primary producers and their transfer across Antarctic marine food webs. In fact, the underside of sea-ice is a unique habitat for well-developed communities of algae which are important food resources for grazing marine organisms [98]. Antarctic krill is the main food source for many organisms, and while juvenile krill protects itself from predators and currents by sheltering in small ridges and crevices of the sea-ice, larvae of *E. superba* scrape algae from the underside of ice to survive the winter [99].

## 4. Local Contamination by MPs in Coastal Ecosystems

A preliminary survey on the occurrence of MPs carried out in 2016 throughout the Southern Ocean reported much higher concentrations (about 100,000 items km^−2^) in two stations close to the continent than in offshore ones [81]. It was speculated that MPs may have originated from northern inhabited areas, but once transported beyond the Antarctic Circumpolar Current and oceanic fronts, they are likely trapped around Antarctica. This hypothesis needs to be confirmed; however, Jones-Williams et al. [85] also found higher concentrations of micro- and mesoplastics in surface water and in amphipods from the coast of the Antarctic Peninsula compared to samples from open-ocean stations in the sub-Antarctic.

Fibers from cloth washing are among the most common MPs in coastal marine ecosystems affected by scientific stations’ wastewater, and their concentrations in waters and sediments can be comparable to those reported in other coastal environments outside Antarctica. At Admiralty Bay (King George Island) microfibers were found to be entangled in various species of zooplankton [94] and MP concentrations in nearshore waters at South Georgia Island were higher than those reported for other Antarctic coastal environments (Table 1). In summer, King George Island is characterized by fairly intensive scientific, tourism and fishing activities and by analysing anthropogenic microfibers collected from 2012 to 2015 with a sediment trap located at 25 m depth at Potter Cove (Maxwell Bay), downward fluxes were estimated ranging from 115 to 152,750 items m^−2^ day^−1^ [88]. Although comparisons are not meaningful due to different marine environments, these values were much higher than those reported in sediment traps >2000 m depth in the subtropical North Atlantic (mean microfiber flux rate = 94 items m^−2^ day^−1^) [100]. Anyhow, this study seems to indicate that most MPs released by scientific stations accumulate in coastal marine sediments. Up to five particles were found in 10 mL of sediment near the sewage treatment plant outfall at Rothera Station [101] and Zhang et al. [82] found higher MP abundance in surface waters of the Ross Sea and in subsurface samples from the Dumont d’Urville Sea, where the French Station is located. The Ross Sea is one of the largest marine protected areas worldwide and plastic debris was quite widespread in sediments collected near the Italian scientific station Mario Zucchelli [102]. In the same coastal area, textile fibers, mainly from technical clothing of the station personnel, were found in 27.3% of Antarctic whelk (*Neobuccinum eatoni*) [103], and in most of 12 macrobenthic species collected from three sites at increasing distance from the station [104] (Table 2). The particles were mainly composed of nylon and polyethylene (mostly 50–100 µm in size) and their concentrations (0.01 to 3.29 items mg^−1^) were three to five times higher in filter-feeders (bivalves) and benthic grazers than in predators or omnivorous invertebrates. Although this accumulation pattern seemed to exclude the transfer of MPs through the Antarctic benthic food web, Bottari et al. [105] have recently found natural and synthetic fibers in the fish *Trematomus bernacchii*, an opportunistic and generalized feeder on benthic organisms, small fishes and zooplankton, collected in 1998 near the Italian station. The similar polymer composition recorded in sediments, benthic organisms and *T. bernacchii* confirms that although the Mario Zucchelli station has a wastewater treatment plant and is only open from mid-October to mid-February, it is a source of local MP contamination. According to Caruso et al. [38] the station accounts for at least 50% of local MP contamination of macrobenthos, while the occurrence of nylon and polyethylene debris also suggests contributions from remote sources, as reported by Fang et al. for Arctic and sub-Arctic macrobenthic communities [106]. Along the Ross Sea coast, other scientific stations have been in operation for decades, such as McMurdo Station (US) and Scott Base (New Zealand), to which the Korean Jan Bogo and the fifth Chinese Antarctic Station have recently been added. The higher anthropogenic impact in this region of the Southern Ocean is consistent with the finding of MPs being more abundant in fishes from the Ross Sea than in those from the Amundsen Sea (where there are no scientific stations) (Table 2).

Apex predators, and especially those consuming their whole prey, act as integrators of MPs ingested by a diversity of species and their scats could reflect the availability of plastic debris in the marine environment. After the discovery in the 1980s of plastic debris having been ingested by seabirds from the Southern Hemisphere [37,112], in recent years some surveys have been carried out on scats of penguins and seals. Bessa et al. [109] analysed 80 scats of gentoo penguins (*Pygoscelis papua*) from Bird Island and Signy Island and identified seven different polymer types in 20% of the samples (Table 2). A higher percentage of microfibers was reported for king penguin (*Aptenodytes patagonicus*) scats at South Georgia (Table 2). Incubating birds, which travel longer distances than chick-rearing penguins to forage further north at the APF, had much higher concentrations of fibers; however, most fibers were of natural cellulosic origin such as cotton or linen [110]. Fragao et al. [111] performed long-term monitoring (2006–2016) in the Antarctic Peninsula and the Scotia Sea of Adélie (*Pygoscelis adeliae*), chinstrap (*Pygoscelis antarcticus*) and gentoo penguin colonies (Table 2). In the three species, the amount of krill and MPs was 85% and 15% in Adélie, 66% and 29% in gentoo, and 54% and 28% in chinstrap penguins, respectively. During the 10-year survey, no temporal variations in MP abundance were observed and it was concluded that significant local sources of plastic contamination exist in the studied areas. However, the emperor penguin (*Aptenodytes forsteri*) is the only penguin species breeding around Antarctica during the austral winter. Golubev [113] reported the ingestion of macroplastic by only one adult emperor penguin and a portion of this material being subsequently used for feeding a chick, whereas Leistenschneider et al. [108] found no MP ingestion in emperor penguin chicks at Atka Bay (Dronning Maud Land) (Table 2). They found 85 putative particles in 41 chicks; however, analysis using Attenuated Reflection Fourier-transform Infrared (ATR-FTIR) spectroscopy showed that none of the particles were MPs, but fibers originating from contamination during sample processing and analysis. Thus, in contrast with previous studies on penguins, it was stated that in continental Antarctica there is no trophic transfer and biomagnification of MPs in the emperor penguin and the need was stressed for standardized procedures in processing and analysing MPs in scat samples.

The occurrence of plastic particles in scats of the Antarctic fur seal (*Arctocephalus gazella*) on Macquarie Island (north of the APF) has been reported since 2003 [114]. More recently, Perez-Venegas et al. [115,116] found microfibers in scats from a population of South American fur seals (*A. australis*) in Northern Patagonia and plastic fragments and fibers also in scats from different species of pinnipeds living on the Peruvian and Chilean coasts. However, as reported for emperor penguins at Atka Bay [108], Garcia-Garin et al. [117] analysed with ATR-FTIR spectroscopy the fibers and fragments found in 42 scats of male Antarctic fur seals from Deception Island (South Shetland Islands; a hotspot of human activities in Antarctica), and found silicate minerals and chitin fibers, but no MPs.

The reviewed results indicate that the methods for extracting and characterizing plastic debris from environmental matrices are still inadequate and, until the technical and methodological challenges are resolved and standardized, comparisons between different surveys will not be fully reliable. Most of the studies have focused on the largest plastic debris floating on the sea surface, which is exposed to the action of UV rays, wind and ice; but many small particles resulting from its fragmentation are found in the water column, as well as fibers deposited from local and long-range atmospheric transport, which cannot be collected with the commonly used mesh size [118]. However, the data acquired so far seem to indicate that, with the exception of some coastal areas directly affected by wastewater from scientific stations and other human activities, MPs are expected to have negligible or undetectable direct biological effects throughout the Southern Ocean. Therefore, it would seem appropriate to focus attention on the most-impacted coastal ecosystems and on the possible interactions of MPs with other anthropogenic contaminants in the context of predicted climate and environmental changes.

## 5. Biotic Interactions of MPs

Although few diatom taxa, such as *Fragillariopsis*, *Thalassionema* and *Eucampia*, support the food web and play a fundamental role in the functioning of marine ecosystems in the Southern Ocean, their interactions with plastic debris have been poorly investigated [119,120]. Some field studies and microcosm experiments show that MPs are more concentrated within sea-ice than in underlying seawater [97,121]; however, the mechanisms by which MPs are captured in sea-ice are unknown. A laboratory experiment by Hoffmann et al. [119] indicates that the exopolymeric substances produced by ice algae can influence the surface binding properties of MPs during sea-ice growth. Interactions between sea-ice algae and MPs also appear to be supported by the finding that in an Antarctic sea-ice core, MP concentrations were positively correlated with those of chlorophyll α [122]. Sea-ice algae are essential for the wintering of young Antarctic krill (age class 0) [123], and their release during the summer melting of sea-ice coincides with the bloom of sympagic and pelagic communities at the ice-edge [124]. Smaller plastic particles, and especially those less dense than seawater such as polyethylene or polypropylene fragments, can be absorbed by phyto- and zooplankton organisms. Exposure experiments show that Antarctic krill ingests and egests polyethylene microspheres without bioaccumulation and acute toxicity [125] and can fragment ingested plastic particles of 31.5 µm into pieces less than 1 µm [126]. The formation of nanoplastics could increase the bioavailability and potential biological impact of MPs. Bergami et al. [127], for instance, found impairments in moulting and swimming and in the excretion of nanoplastics via fecal pellets in juveniles of *E. superba* exposed to plastic nanospheres (50–60 nm).

Ice scouring severely limits the extent and biodiversity of biotic communities on the Antarctic shores; however, starting from depths of 3–5 m, most of substrata are covered by a vertical succession of many different macroalgae, with the genera *Palmaria*, *Desmaretia* and *Himanthothallus* reaching down to 90–100 m depths [128]. Macroalgae can absorb MPs directly from water and/or through the polysaccharide colloidal layer coating their surface [129]. Thus, in the Antarctic coastal environment and particularly in bays and fjords, where most scientific stations are located, macroalgae can play an important role in the transfer of MPs and other contaminants to marine invertebrates, either through direct consumption or via detritus.

Due to their persistent and hydrophobic nature, buoyant plastic fragments of a few millimeters or hundreds of micrometers can be colonized by microalgae and many other organisms such as viruses, bacteria, fungi and invertebrates (e.g., cnidarians, bryozoans or barnacles) [130,131]. Epiplastic communities modify the density and the vertical flux of plastic debris and the binding properties of their surfaces. Just like paint fragments, MPs with biofouling or attached to marine snow, faecal pellets and zooplankton organisms settle in the sedimentary environment where they can be ingested by benthic fauna [94]. On the surface of plastic and paint fragments collected around the Antarctic Peninsula there were coccoid and filamentous bacteria, microalgae and some invertebrate species [84]. A wide range of phylogenetically diverse bacteria have also been found on plastic debris, including pathogenic species [131]. Prokaryotic communities on a marine plastic fragment from King George Island were dominated by Gamma- and Betaproteobacteria [132]; moreover, on the same island, bacteria strains with multiple antibiotic resistance were isolated from a beached polystyrene fragment [133]. Thus, in the same way as small pieces of wood or other floating materials, plastic debris can introduce alien and invasive species of fouling organisms into the Southern Ocean and/or spread them through different Antarctic marine regions.

## 6. Future Climate and Environmental Scenarios

As reported above, sea-ice intercepts and accumulates MPs from atmospheric deposition and seawater and probably contributes to their ad/absorption by primary producers and Antarctic krill [119,121,129]. About 15 million km^2^ of seasonal sea-ice grow and melt around Antarctica each year; however, this extensive sea-ice cover shows wide inter-annual variations. After a few years of great extension, a rapid decline in Antarctic sea-ice has begun since 2016 (the most pronounced since the beginning of 40 years of satellite observations, corresponding to 30 years of sea-ice loss in the Arctic) [134]. This macroscopic reduction is likely due to progressive ocean warming and the southward advection of atmospheric heat, which in turn can affect the productivity of the Southern Ocean, and the long-range transport of MPs. Climate stress and MPs can also influence the physiological processes of Antarctic marine organisms themselves, which tolerate minimal temperature variations. Kratina et al. [135], for instance, tested the independent and combined impact of MPs and water temperature on a freshwater detritivore amphipod and found that the MP exposure could alter its metabolic rate, with greater inhibition at higher temperatures.

The absorption of anthropogenic CO_2_ by the oceans is lowering seawater pH and carbonate ion concentrations. Due to the higher solubility of CO_2_ at lower temperatures, naturally low concentrations of calcium carbonate and the upwelling of CO_2_-rich deep waters, Antarctic marine organisms are likely to be most exposed to the potentially damaging effects of seawater acidification [136]. Calcifying organisms depend on aragonite saturation in seawater and it has been estimated that those in the Southern Ocean could experience aragonite undersaturation by 2050. Not surprisingly, Negrete-Garcia et al. [137] suggested that the Southern Ocean waters could act as a “bellwether” of water acidification in the global ocean. Despite the sensitivity of the Southern Ocean to global warming and water acidification, there are few studies on their possible effects on Antarctic marine organisms. The results of some preliminary laboratory studies seem to indicate that high CO_2_ concentrations increase the nutrient uptake and growth of Antarctic sea-ice diatoms of the genus *Nitzschia* [138]; low pH values do not appear to influence stress responses by the Antarctic limpet *Nacella concinna* [139]; and the fertilization of the Antarctic echinoid *Sterechinus neumayeri* appears to be resilient to the near-future ocean warming and acidification [140]. Adult Antarctic krill, *E. superba*, also appears not to be affected by exposure for one year to near-future levels of ocean acidification (1000–2000 µatm ρCO_2_) [141]. However, other studies suggest that regardless of MPs, Antarctic krill is sensitive to environmental changes due to rising water temperature and sea-ice loss [142], and Kawaguchi et al. [143] found an inhibition of embryonic development of *E. superba* at a simulated concentration of 2000 µatm ρCO_2._ Biological effects were also observed in experiments with multi-stress treatments. Rowland et al. [144] found a lower proportion of developing *E. superba* embryos when exposed to the combined effects of different nanoplastics and water acidification and this treatment had negative effects on the survival of the sub-Antarctic pteropod *Limacina retroversa*, even with short-term exposure (48 h) [145]. Since the biological thresholds of any stressors can be influenced by the concomitant effects of other stressors, the latter two ecotoxicological studies have concluded that to assess the impact of micro- and nanoplastics in Antarctic organisms it is necessary to consider future climate scenarios and environmental changes.

## 7. MP Interactions with Other Contaminants and Possible Cumulative Stress

Ingested MPs particles can act as vectors for microorganisms and persistent organic pollutants, and the concomitant impact of pathogens and chemicals is undoubtedly far more severe than internal abrasion or other direct effects of MPs alone. As early as 1998, Ryan et al. [146], analyzing the mass of ingested plastic and concentrations of PCBs, DDT, dichlorodiphenyldichloroethylene (DDE) and dieldrin in the fat and eggs of breeding females of the great shearwater (*Puffinus gravis*), found a positive relationship between plastic and PCB content and hypothesized that the ingestion of plastic debris may contribute to the absorption of toxic chemicals by seabirds. Micro- and nanoplastics have a large specific surface and, depending on the characteristics of the polymers, their surfaces can provide an ecological niche for microorganisms with the formation of biofilms (called the plastisphere), which can enhance the adsorption of metals and organic pollutants [147,148,149].

One main finding in MP research in surface and sub-surface waters of the Southern Ocean was the widespread occurrence of paint fragments derived from vessels. These particles can adsorb contaminants and, if ingested, can also release toxic chemicals in paint and dyes [83,84], which have already been detected in phytoplankton samples from Deception Island [93]. Moreover, Brennecke et al. [48] found that the metals released by antifouling paint into the water were efficiently adsorbed by virgin polystyrene beads and aged polyvinyl chloride fragments. Therefore, in some coastal areas of the Antarctic with rather intensive human activities, the concomitant occurrence of MPs and paint debris could enhance the bioavailability and accumulation of potentially toxic metals in pelagic and benthic organisms.

Several kinetic models have been used to study the adsorption of metals and POPs by MPs, and among the parameters tested (e.g., temperature, pH, contact time, ionic strength) there is evidence that the aging of particles, by increasing their specific surface and the oxygen-containing functional groups, plays an important role [148]. Being used by algae as a substitute for Zn, high concentrations of Cd accumulate in the livers and kidneys of Antarctic marine organisms [55,56,57], and it has been shown that some polymers, such as polyvinylchloride, polystyrene and polyethylene terephthalate, have a high adsorption capacity for metals such as Cd and POPs. This capacity is due to the presence on their surface of functional groups such as polar atomic chlorine, phenyl and ester groups and carboxyl groups formed by photodegradation [150]. The concentrations of metals, organic pollutants and microorganisms in MPs can be orders of magnitude higher than in seawater and upon MP ingestion they can be desorbed and assimilated in the organs and tissues of marine organisms [151].

Regardless of the relative importance of MPs as carriers of contaminants for the biota compared to other absorption routes, in Antarctic coastal marine ecosystems near scientific stations, chronic co-exposure even to low concentrations of a range of contaminants and pathogenic microorganisms can give rise to synergistic or additive biological effects. Most personal Antarctic equipment is made from synthetic polymers, which are often treated with water-repellent and flame-retardant compounds. Thus, wastewater from scientific stations is a source of these contaminants and of fibers from clothing, chemicals from detergents, personal care products or pharmaceuticals, and pathogenic microorganisms [14,20,25,26,27]. Through the displacement of essential microbial species regulating physiological functions, pathogenic microorganisms can cause dysbiosis in the gastrointestinal tract of fish, with inflammatory processes and possible interference with the immune system and chronic diseases [152]. Under stress conditions, such as those due to the enhanced bioavailabilty of Cd, Hg and Zn, bacteria can become resistant to antibiotics [152] and strains with multiple-antibiotic resistance have already been isolated in a polystyrene fragment beached on King George Island [133].

The biological implications for Antarctic organisms from simultaneous exposure to MPs, metals, organic compounds and microorganisms remain to be explored. Moreover, the resilience and responses of communities and ecosystems could be exacerbated by the concomitant impact of climate-related stressors and other anthropogenic disturbances. Just think of the pressures on Antarctic krill, the keystone species of the Southern Ocean trophic web, with the largest biomass (300 to 500 million tonnes) of all multicellular organisms. The success of *E. superba* is due to species-specific physiological and behavioral characteristics, such as adaptation to the Southern Ocean environment, synchronization with the seasonal sea-ice cycle and the ability to exploit whatever food is available. The huge decrease in Antarctic sea-ice in recent years is likely giving krill less food and shelter from predation, and many other threats are putting the conservation of *E. superba* populations at risk. For over a century, krill biomass has declined in parts of the Atlantic sector of the Southern Ocean (a region more than half of krill populations, large colonies of penguins and other seabirds, and many marine mammals inhabit). The fishing industry and krill catches are concentrated around the South Orkney Islands and the Antarctic Peninsula and the increasing demand for fish feed and for the production of pharmaceuticals and nutraceuticals from krill could increase the pressure on the krill fishery, placing further threats on this fundamental resource for Southern Ocean food webs [153].

## 8. Conclusions

The Antarctic continent and the Southern Ocean are linked to global processes and receive anthropogenic persistent contaminants released worldwide through long-range atmospheric and marine transport. Moreover, the number of persons visiting Antarctica for scientific, tourism and fishing activities is increasing and can directly affect the Antarctic environment through fuel combustion, accidental oil spills, sewage outfalls, waste production and the introduction of pre-adapted and invasive alien species. Plastic debris is one of the most pervasive and ubiquitous synthetic wastes in the global environment and in recent years there has been growing concern about its occurrence in Antarctic marine ecosystems. However, despite the extraordinary increase in the number of surveys, it is still impossible to establish the distribution of MPs in the Southern Ocean and their potential impact on biotic communities. Overall, plastic contamination appears to be significant, especially in coastal marine areas of the Antarctic Peninsula, sub-Antarctic islands and the Ross Sea that receive wastewater from scientific stations and/or with rather intense research, fishing or tourism activities. The Southern Ocean has specific biotic communities and food webs and to evaluate the possible biological effects of MPs it is advisable to direct future research towards:-the standardization of sampling and analytical procedures, paying greater attention to nanoplastics and adopting universally accepted methods for the assessment of MP contamination of samples in the field and in the lab.-the study of possible interactions of MPs with macro- and microalgae (especially sea-ice algae) for their potential role in the transfer of contaminants to Antarctic krill and along the food web.-the adsorption on plastic debris of potentially toxic anthropogenic contaminants and the composition of epiplastic biotic communities, which can be sources of pathogenic microorganisms.-the responses of keystone species under environmentally realistic conditions to combined exposure to MPs, paint chips, metals, POPs and CEI, which are quite common in Antarctic coastal ecosystems impacted by human activities.

In fact, these cumulative stresses could likely reduce the resilience of Antarctic marine organisms and ecosystems, thereby increasing the impact of anthropogenic contaminants and pathogenic microorganisms.

Regarding the implementation of environmental management protocols, it seems appropriate to raise awareness of all people visiting Antarctica about the potential impact of MPs, personal care products, pharmaceuticals and inadvertently introduced alien organisms. National Antarctic Research Programs and tour operators should be encouraged to equip scientific stations and vessels with suitable wastewater treatment facilities to reduce/prevent the discharge of MPs, contaminants and microorganisms into Southern Ocean waters.

## Figures and Tables

**Table 1 animals-13-00162-t001:** Indicative values of microplastic abundance (obtained with different sampling and analytical methods) in seawaters from the Southern, Arctic and Atlantic Oceans.

	Seawater	Size (Range)	Item Abundance (Mean or Range)	Ref.
**Southern Ocean**				
Antarctic circumnavigation	Surface	0.2–25 mm	353 km^−2^	[33]
Antarctic circumnavigation	Surface	>300 μm	No MPs. (0.002–1.336 m^−3^ items of biogenic origin)	[80]
East Antarctica	Surface	0.35–5.5 mm	0.046–0.099 m^−3^	[81]
East Antarctica	Surface Subsurface (8 m)	>330 μm	0.006–0.44 m^−3^ (0.10 ± 0.14) 0.13–4.41 m^−3^ (1.69 ± 1.21)	[82]
Weddell Sea	Surface Subsurface (11.2 m)	0.1–8.7 mm 0.1–1.4 mm	0.01 ± 0.01 m^−3^ 0.04 ± 0.1 m^−3^	[83]
Ross Sea	Subsurface (5 m)	>60 μm	0.17 ± 0.34 m^−3^	[32]
Ross Sea	Surface Subsurface (8 m)	>330 μm	0.1–0.5 m^−3^ 2.0–4.0 m^−3^	[82]
West Antarctic Peninsula	Surface	0.5–75 mm	0.008 m^−3^ 755–3524 km^−2^	[84]
Mid Scotia Sea- AntarcticPeninsula	Surface	0.16–10 μm	0.013 ± 0.005 m^−3^	[85]
King George Island	Entangled in zooplankton (from 0 to 30 m depth)	>150 μm	2.40 ± 4.57 100 m^−3^	[86]
South Georgia Island	Nearshore (<1 m) Station wastewater	>55 μm	1.75 ± 5.17 L^−1^ 1.66 ± 3.00 L^−1^	[87]
King George Is. Potter Cove	Sediment trap At 25 m depth	10–450 μm	Estimated microfiber flux: 115–152,750 m^−2^	[88]
Scotia Sea Antarctic Peninsula	Subsurface (3 m)	0.1–1.5 mm	0.43 m^−3^	[79]
**Arctic Ocean**				
Transect through the Arctic Ocean	Subsurface (3–8 m)	>63 μm	40.5 ± 4.4 m^−3^	[89]
North Pole	Subsurface (3–8) m	>63 μm	44.3 ± 6.9 m^−3^	[89]
Siberian Arctic	Subsurface (3 m)	0.1–5 mm	0.71 m^−3^	[79]
From East Asian seas to Arctic central basin	Subsurface (8 m)	>300 μm	2.91 ± 1.93 m^−3^	[90]
Beaufort Sea	Deep (up to 1015 m)	>63 μm	37.3 ± 6.9 m^−3^	[89]
Barents Sea	Subsurface (3 m)	0.1–1.5 mm	0.85 m^−3^	[79]
**Atlantic Ocean**				
Central Atlantic	Subsurface (3 m)	0.1–5 mm	0.78 m^−3^	[79]
Between 32° N and 32° S	Subsurface (3–8 m)	>200 μm	1.15 ± 1.45 m^−3^	[91]

**Table 2 animals-13-00162-t002:** Occurrence of plastic debris in benthic invertebrates, fish and penguin scat from continental Antarctica, the Antarctic Peninsula and the Scotia Arc.

	Marine Area	Species	Type and n of Items	Ref.
**Continental** **Antarctica**				
Ross Sea	Near Mario Zucchelli Station	*Neobuccinum eatoni*	Microfibers (length 0.8–5.7 mm); in 27.3% of whelks	[103]
Ross Sea	Near Mario Zucchelli Station	Macrobenthic invertebrates (12 species)	Length: 33–1000 μm; in 83% of specimens; 0.4–1.9 items individual^−1^	[104]
Ross Sea	Near Mario Zucchelli Station	*Trematomus bernacchii*	Fibers 95%, fragments 5%; size range 0.4–4.2 mm; in 75% of analysed fishes	[105]
Ross Sea	Off shore depth 285–332 m	10 fish species from 5 families	Mostly polyacrylamide (PAM); length 100–200 μm; in 50% of specimens; 1.286 items individual^−1^	[107]
Amundsen Sea	Off shore 522–659 m	10 fish species from 5 families	Mostly rayon; length 500–1000 μm; in 36% of specimens; 1.227 items individual^−1^	[107]
Queen Maud Land Atka Bay	Coastal sea-ice	*Aptenodytes forsteri* (scat)	No MPs. Only fibers from contamination of samples	[108]
**Scotia Sea and** **Antarctic Peninsula**				
Scotia Sea	Bird Island Signy Island	*Pygoscelis papua* (scat)	Mostly polyester fibers 58%; fragments 26%; films 16%; size from 76 to 4945 μm; in 20% of samples; 0.23 ± 0.53 items individual scat^−1^	[109]
Scotia Sea	South Georgia	*Aptenodytes patagonicus*	Microfibres 77%; 186–9280 μm in length; 21.9 ± 5.8 microfibres g^−1^ of faeces; synthetic MPs 12%	[110]
Antarctic Peninsula and Scotia Sea	Yalour Island Deception Island Hannah Point Rongé Island	*Pygoscelis adeliae*	Mostly polyethylene (80%); fibres 74%, fragments 44%; size 63–1000 μm; in 26% of scats; 0.15 ± 0.4 scat^−1^	[111]
	King George Is. Paradise Bay King George Is. Hannah Point Rongé Island Cierva Cove	*Pygoscelis antarcticus*	in 29% of scats; 0.31 ± 0.5 scat^−1^	
	King George Island Bird Island Byers Peninsula	*Pygoscelis papua*	in 29% of scats; 0.29 ± 0.5 scat^−1^	

## Data Availability

Not applicable.

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
