# Peer review of "Microplastic Interactions and Possible Combined Biological Effects in Antarctic Marine Ecosystems"

_animals, 2022, doi:10.3390/ani13010162_

Round 1

Reviewer 1 Report

I revised the manuscript “The Effects of Microplastics on Marine Biota” submitted to the Animals.  

The authors presented the topic on the microplastic in marine ecosystem. 

The interactions of MPs with other anthropogenic contaminants and on possible cumulative stress due to predicted changes in the Antarctic climate and marine environment is discussed.

 Furthermore, authors currented approaches to microplastic in evironmental are introduced.  The topic of the article is up to date, the introduction is easy detailed. This problem is relevant for journal scope. 

The concept and aim are clearly defined.  The presentation and discussion of the presented topicis clear and very detailed.

The authors discussed almost all available literature sources.

Other weaknesses to be corrected:

1. Keywords should be in alphabetical order.

2. I propose to insert a dictionary for abbreviations

3. The conclusions should be shorter and more specific. I propose to shorten the conclusions and keep the most important ones. It is best to list 3-4 critical conclusions.

The manuscript follows the formal regulations of MDPI journals.

I suggest the acceptance after minor revision

Author Response

Many thanks for your thoughtful and constructive review. We followed all your suggestions:

Keywords are now listed alphabetically. A list of abbreviations is provided in Appendix A. The conclusions have been shortened and are now more specific.

Roberto Bargagli & Emilia Rota

Reviewer 2 Report

Dear Authors,

thank you for you valuable article. I need to ask for some corrections:

- please try to shorthen the abstract and maybe please decrease the number of keywords

- line 73, 117, 119, 130, 152 - please remove e.g. before the number of references,

- line 365: please check unit m-2/day or m2/day

Author Response

Many thanks for your thoughtful and constructive review. Some English errors have been corrected. The abstract has been shortened, the keywords are now more concise. Other spelling errors have been checked and corrected.

Roberto Bargagli & Emilia Rota

Reviewer 3 Report

This manuscript addresses the potential risks of microplastics to marine flora and fauna in the Southern Ocean and coastal waters off the Antarctic Peninsula and reviews studies conducted to date to quantify inputs and effects. Although these habitats are isolated from dense population centers and industrial operations, inputs of microplastics can still be derived from scientific stations, fishing, and tourism activities. Lack of standardized methodologies for quantifying inputs and effects of microplastics has resulted in inadequate assessments of potential risks, especially in assessing populations at risk such as Emperor penguins, and the changing environmental conditions.  In addition, the paper highlights the effects of human activities on vulnerable species and ecosystems and how the Antarctic Treaty’s Protocol on Environmental Protection (ATPEP) can be implemented to support environmental management.

The authors provide an extensive review of the scientific literature pertaining to habitat surveys, presence of persistent chemicals (including microplastics, and potential impacts in different regions of the Southern, Arctic, and Atlantic Oceans. Unlike the other oceans, the Southern Ocean has no riverine input and, thus, sources of persistent contaminants are derived from land-based operations, scientific research vessels, fishing vessels, or cruise ships. The authors conclude that even in coastal ecosystems that would be most likely impacted by local sources of contaminants, direct effects of microplastics on organisms is most likely to be minimum or undetectable. Yet, there is a need for standardized approaches to characterize exposure, effects, and interactions of microplastics and nanoplastics within biota in the Antarctic environment, and potential combined effects with other stressors and transfer within marine food webs. Combined effects exposure to microplastics and other stressors (e.g., climate variables, ocean acidification, and interactions among contaminants), and practices to reduce inputs of microplastics and other contaminants to the Southern Ocean warrant further investigation.

This paper will be a valuable contribution to the special volume on "The Effects of Microplastics on Marine Biota".

Author Response

Thank you very much for your careful review and valuable comments.

Roberto Bargagli & Emilia Rota